# Gene-environment interaction in molar-incisor hypomineralization

**Mariana Bezamat**[1], **Juliana F. Souza**[2], **Fernanda M. F. Silva**[3], **Emilly G. Corrêa**[4], **Aluhe L. Fatturi**[2], **João A. Brancher**[5], **Flávia M. Carvalho**[6], **Tayla Cavallari**[4], **Laís Bertolazo**[4], **Cleber Machado-Souza**[7], **Mine Koruyucu**[8], **Merve Bayram**[9], **Andrea Racic**[1], **Benjamin M. Harrison**[1], **Yan Y. Sweat**[10], **Ariadne Letra**[11], **Deborah Studen-Pavlovich**[12], **Figen Seymen**[8], **Brad Amendt**[10], **Renata I. Werneck**[4], **Marcelo C. Costa**[3], **Adriana Modesto**[12], **Alexandre R. Vieira**[1]*

1 Department of Oral Biology, University of Pittsburgh, Pittsburgh, Pennsylvania, United States of America, 2 Department of Stomatology, Federal University of Paraná, Curitiba, State of Paraná, Brazil, 3 Department of Pediatric Dentistry and Orthodontics, Federal University of Rio de Janeiro, Rio de Janeiro, Rio de Janeiro, Brazil, 4 Graduate Program of Dentistry, Pontifical Catholic University of Paraná, Curitiba, State of Paraná, Brazil, 5 Graduate Program of Dentistry, Positivo University, Curitiba, State of Pará, Brazil, 6 Department of Genetics, Oswaldo Cruz Foundation (FIOCRUZ), Rio de Janeiro, Rio de Janeiro, Brazil, 7 Graduate Program of Applied Biotechnology to Child and Adolescent Health, Pequeno Príncipe College, Curitiba, State of Pará, Brazil, 8 Department of Pedodontics, Istanbul University, Istanbul, Turkey, 9 Department of Pedodontics, Medipol Istanbul University, Istanbul, Turkey, 10 Craniofacial Anomalies Research Center and Department of Orthodontics, College of Dentistry, The University of Iowa, Iowa City, Iowa, United States of America, 11 Department of Diagnostic and Biomedical Sciences, and Center for Craniofacial Research, UTHealth School of Dentistry at Houston, Houston, Texas, United States of America, 12 Department of Pediatric Dentistry, University of Pittsburgh, Pittsburgh, Pennsylvania, United States of America

* arv11@pitt.edu

## Abstract

Molar incisor hypomineralization (MIH) is an enamel condition characterized by lesions ranging in color from white to brown which present rapid caries progression, and mainly affects permanent first molars and incisors. These enamel defects usually occur when there are disturbances during the mineralization or maturation stage of amelogenesis. Both genetic and environmental factors have been suggested to play roles in MIH's development, but no conclusive risk factors have shown the source of the disease. During head and neck development, the interferon regulatory factor 6 (*IRF6*) gene is involved in the structure formation of the oral and maxillofacial regions, and the transforming growth factor alpha (*TGFA*) is an essential cell regulator, acting during proliferation, differentiation, migration and apoptosis. In this present study, it was hypothesized that these genes interact and contribute to predisposition of MIH. Environmental factors affecting children that were 3 years of age or older were also hypothesized to play a role in the disease etiology. Those factors included respiratory issues, malnutrition, food intolerance, infection of any sort and medication intake. A total of 1,065 salivary samples from four different cohorts were obtained, and DNA was extracted from each sample and genotyped for nine different single nucleotide polymorphisms. Association tests and logistic regression implemented in PLINK were used for analyses. A potential interaction between *TGFA* rs930655 with all markers tested in the cohort from Turkey was identified. These interactions were not identified in the remaining cohorts. Associations (p<0.05) between the use of medication after three years of age and

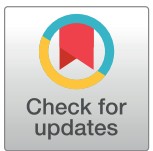

**Data Availability Statement:** All relevant data are within the paper and its Supporting information files.

**Funding:** F.M.F.S. was supported by Fundação de Amparo à Pesquisa do Estado do Rio de Janeiro–

FAPERJ process E-26/201.745/2019 (https://www.
google.com/search?source=univ&tbm=isch&q=
Funda%C3%A7%C3%A3o+de+Amparo+%C3%A0
+Pesquisa+do+Estado+do+Rio+de+Janeiro%E2%
80%93+FAPERJ+process+E-26/201.745/
2019&sa=X&ved=2ahUKEwiyt97Vw_XqAhW-
hXlEHbfJCoEQsAR6BAgBEAE&biw=956&bih=
665). E.G.C. was supported by CAPES. A.R. was
supported by the University of Pittsburgh School of
Dental Medicine Dean's Summer Research
program. The work was supported in part by a
grant from Araucária Foundation awarded to R.I.W
(www.fappr.pr.gov.br). This paper is based in part
on a thesis submitted to the graduate faculty,
Federal University of Rio de Janeiro, in partial
fulfillment of the requirements for the PhD degree
(for F.M.F.S.) and on a thesis submitted to the
graduate faculty, Pontifical Catholic University of
Paraná, in partial fulfillment of the requirements for
the MS degree (for E.G.C.). The funders had no role
in study design, data collection and analysis,
decision to publish, or preparation of the
manuscript.

**Competing interests:** The authors have declared
that no competing interests exist.

MIH were also found, suggesting that conditions acquired at the age children start to social-ize might contribute to the development of MIH.

## Introduction

The hardest biological material in the human body is tooth enamel, composed of both mineral and organic phases [1]. Disturbances during amelogenesis impact maturation or mineraliza-tion stages, and can lead to defects in enamel translucence, known as enamel hypomineraliza-tion [2]. Molar-incisor hypomineralization (MIH) is an example of an enamel condition, and is characterized by the appearance of lesions ranging from white to brownish coloration which present rapid caries progression and hypersensitivity [3]. These asymmetrical lesions affect the first permanent molars, usually with the permanent incisors [4], and more recently MIH has been reported to affect canines too [5]. Prevalence of MIH varies but has been consistently reported to range between 1% and 35% in all parts of the world, with most reported frequen-cies around 12% [6–22].

Regarding the timing of these lesions' appearance, reports have shown a variation in the chronology of crown completion and enamel formation of incisors and molars. Gleiser and Hunt documented crown completion of first permanent molars to be between 2.5 and 4.4 years [23]. Boyde reported the chronological age of enamel completion in the upper central incisor as 4.64 years and Reid documented complete crown formation in lower central incisors to be at 3.8 years [24].

The enamel maturation stage is particularly important for tissue development, and a very sensitive stage in terms of hypomineralized enamel dysplasia [25].

Different systematic reviews concluded that most of the previously studied environmental factors that were believed to be risk factors for MIH did not fully explain the disease etiology [5, 26, 27]. In contrast, etiological factors associated with enamel forming genes [2, 28] or in immune response-related genes [28, 29] have been reported to play a role in the condition's onset.

Over the past few years, our group has focused on studying different phenotypes that may be present in the same individuals with the hypothesis that they may be influenced by the same underlying genetic variation [30]. Conditions such as cleft lip and palate, have shown an increased frequency of abnormal tooth sizes and morphology [31]. Furthermore, this work has shown that the frequency of dental anomalies as a consequence of disturbances in dental devel-opment was much higher in individuals born with clefts, indicating that dental phenotypes should be considered an extended phenotype of oral clefts [32]. Later, it was also shown that *IRF6* and *TGFA* interaction might contribute to oral clefts [33], and that genetic associations for MIH can be identified when studied in combination with cleft lip and palate [34]. *IRF6* is a gene involved in the structure formation of the oral and maxillofacial regions and has been documented as responsible for not only influence amelogenesis, but also play a role in root and crown anomalies of first molars in mice [35]. This same study showed that *irf6* conditional knockout mouse presented reduced enamel density preeruption and delayed enamel forma-tion [35]. *TGFA* on the other hand, is an essential cell regulator, acting during proliferation, differentiation, migration and apoptosis. *TGFA* has also been associated with orofacial clefts [36], and a previous study using the case-parent trio design raised the possibility that *IRF6* and *TGFA* interact and lead to orofacial clefts [37], supporting the finding from our group [33].

**Table 1. Breakdown of populations used in the study.**

| Population | Total Sample | MIH Affected | MIH Unaffected | Sex | |
|---|---|---|---|---|---|
| | | | | Male | Female |
| Curitiba (Federal University of Paraná, Brazil) | 356 | 87 | 269 | 187 | 169 |
| Curitiba (Pontifical Catholic University of Paraná, Brazil) | 200 | 100 | 100 | 92 | 108 |
| Rio de Janeiro, Brazil | 174 | 78 | 96 | 101 | 73 |
| Istanbul, Turkey | 355 | 163 | 172 | 171 | 164 |
| Total | 1,065 | 428 | 637 | 551 | 514 |

In light of the aforementioned evidence, it was hypothesized that *IRF6* and *TGFA* interact and contribute to predisposition of MIH. Additionally, it was hypothesized that environmental factors affecting children that were 3 years of age or older could play a role in the etiology of the disease. Those factors included the presence of any respiratory issues, malnutrition, any type of food intolerance, infection of any sort, and the use of any type of medication.

## Results

One thousand and sixty-five individuals from both Brazilian and Turkish cohorts were analyzed. All four cohorts are described in detail in the (S1–S4 Files) and a breakdown of the populations included in the study is described in Table 1. All markers were in Hardy-Weinberg equilibrium (PLINK threshold p-value bellow 0.001), with the exception of rs930655 in the sample from Curitiba (Federal University of Paraná), that was excluded from further analysis (Table 2). It was also excluded rs2073487, rs642961 and rs2902345 in the sample from Rio de Janeiro, and rs1523305 and rs2902345 in the sample from Curitiba (Pontifical Catholic University of Paraná). The numbers in bold in Table 2 highlight the distribution of the genotypes deviating from Hardy-Weinberg equilibrium.

### Single marker analysis

No significant associations between the selected SNPs and MIH were detected in the study populations.

### Gene-gene interaction analyses

The cohort from Rio de Janeiro showed a trend toward statistical evidence of interaction between *TGFA* rs1523305 and *IRF6* rs642961 (p = 0.03, OR = 1.77) and between *IRF6* rs2073487 and *TGFA* rs2902345 (p = 0.04, OR = 1.60). Significant results (p<0.0004) were found for the cohort from Istanbul between *TGFA* rs930655 and all *IRF6* markers (Table 3).

### Gene expression analysis

It was investigated the localization of Tgfa and Irf6 in wild type mice at critical stages of dental development. It was observed expression of Irf6 in ameloblasts, apparently more membrane-bound than cytoplasmic or nuclear, whereas expression of Tgfa was not observed (Fig 1).

### Environmental factors

Associations were identified between MIH and any type of medications taken at three years of age (Table 4). This variable was designed to capture medication used early in life. In this analysis MIH was used as the dependent variable and medication intake as the independent variable. In the Curitiba cohort (Federal University of Paraná), out of 191 individuals who

**Table 2. Characteristics of the selected variants and genotyping frequencies.**

| Gene | SNP marker | Base change | Consequence | Population | Genotyping Calls | |
|---|---|---|---|---|---|---|
| | | | | | MIH Affected | MIH Unaffected |
| IRF6 | rs2073487 | T>C | Intron variant | Curitiba (Federal University of Paraná) | 14/40/32 | 52/118/88 |
| | | | | Curitiba (Pontifical Catholic University of Paraná) | 20/43/28 | 15/42/38 |
| | | | | Rio de Janeiro | **5/6/55** | **11/11/56** |
| | | | | Istanbul | 15/56/75 | 21/70/80 |
| | rs2013162 | A>C | Synonymous variant | Curitiba (Federal University of Paraná) | 16/41/29 | 51/121/88 |
| | | | | Curitiba (Pontifical Catholic University of Paraná) | 12/63/21 | 11/49/35 |
| | | | | Rio de Janeiro | 4/14/33 | 2/10/50 |
| | | | | Istanbul | 16/54/73 | 23/68/81 |
| | rs17015215 | C>T | Missense variant | Curitiba (Federal University of Paraná) | 0/21/62 | 5/38/209 |
| | | | | Curitiba (Pontifical Catholic University of Paraná) | Not genotyped | |
| | | | | Rio de Janeiro | 6/18/30 | 5/33/24 |
| | | | | Istanbul | 2/3/130 | 1/7/144 |
| | rs861019 | A>G | Intron variant | Curitiba (Federal University of Paraná) | 15/37/35 | 41/120/97 |
| | | | | Curitiba (Pontifical Catholic University of Paraná) | 11/42/31 | 11/49/26 |
| | | | | Rio de Janeiro | 9/35/13 | 10/33/15 |
| | | | | Istanbul | 34/70/42 | 38/69/42 |
| | rs642961 | A>G | None | Curitiba (Federal University of Paraná) | 4/21/61 | 7/70/184 |
| | | | | Curitiba (Pontifical Catholic University of Paraná) | 8/23/65 | 7/21/68 |
| | | | | Rio de Janeiro | 3/39/31 | **2/53/33** |
| | | | | Istanbul | 5/43/99 | 5/36/105 |
| TGFA | rs2902345 | T>C | Intron variant | Curitiba (Federal University of Paraná) | 10/42/30 | 59/102/90 |
| | | | | Curitiba (Pontifical Catholic University of Paraná) | **20/7/52** | **21/12/47** |
| | | | | Rio de Janeiro | **19/15/36** | 11/19/45 |
| | | | | Istanbul | 42/62/47 | 38/60/54 |
| | rs2166975 | G>A | Synonymous variant | Curitiba (Federal University of Paraná) | 2/29/56 | 20/72/164 |
| | | | | Curitiba (Pontifical Catholic University of Paraná) | 7/19/74 | 8/20/71 |
| | | | | Rio de Janeiro | Not genotyped | |
| | | | | Istanbul | 9/38/98 | 7/62/98 |
| | rs1523305 | C>T | Intron variant | Curitiba (Federal University of Paraná) | 13/51/22 | 66/106/86 |
| | | | | Curitiba (Pontifical Catholic University of Paraná) | **24/8/53** | **28/10/53** |
| | | | | Rio de Janeiro | 9/21/35 | 7/22/61 |
| | | | | Istanbul | 43/57/43 | 40/62/46 |
| | rs930655 | A>G | Intron variant | Curitiba (Federal University of Paraná) | 11/31/44 | **61/60/136** |
| | | | | Curitiba (Pontifical Catholic University of Paraná) | 18/48/32 | 10/50/34 |
| | | | | Rio de Janeiro | Not genotyped | |
| | | | | Istanbul | 9/39/98 | 9/54/82 |

Bold indicates figures deviating from Hardy-Weinberg.

reported having taken medications, 52 were affected by MIH and 139 were not. In contrast, out of 166 individuals who reported not taking any medications, 36 were affected with MIH and 130 were not. In the cohort from Rio de Janeiro, 35 individuals reported having taken medications and were affected by MIH, 15 reported having taken medications and were not affected by MIH, 43 did not take any medications and were affected with MIH and 81 did not take medications and were not affected by MIH. In both groups it was found statistical

**Table 3. Summary results of gene-gene interaction analysis between *TGFA* rs930655 and the other *IRF6* markers in the samples from Istanbul.** The odds of the less common allele of *TGFA* rs930655 in addition to the *IRF6* marker allele to increase the chance of MIH to occur.

| SNP | Allele | Prevalence Ratio (95% confidence interval) | p-value |
|---|---|---|---|
| rs17015215 | T | 1.33 (1.25–1.42) | 2.603e-016 |
| rs2013162 | A | 1.25 (1.16–1.34) | 2.992e-019 |
| rs861019 | A | 1.34 (1.26–1.42) | 1.379e-016 |
| rs2073487 | C | 1.24 (1.16–1.34) | 8.509e-019 |
| rs642961 | A | 1.39 (1.31–1.47) | 1.842e-015 |

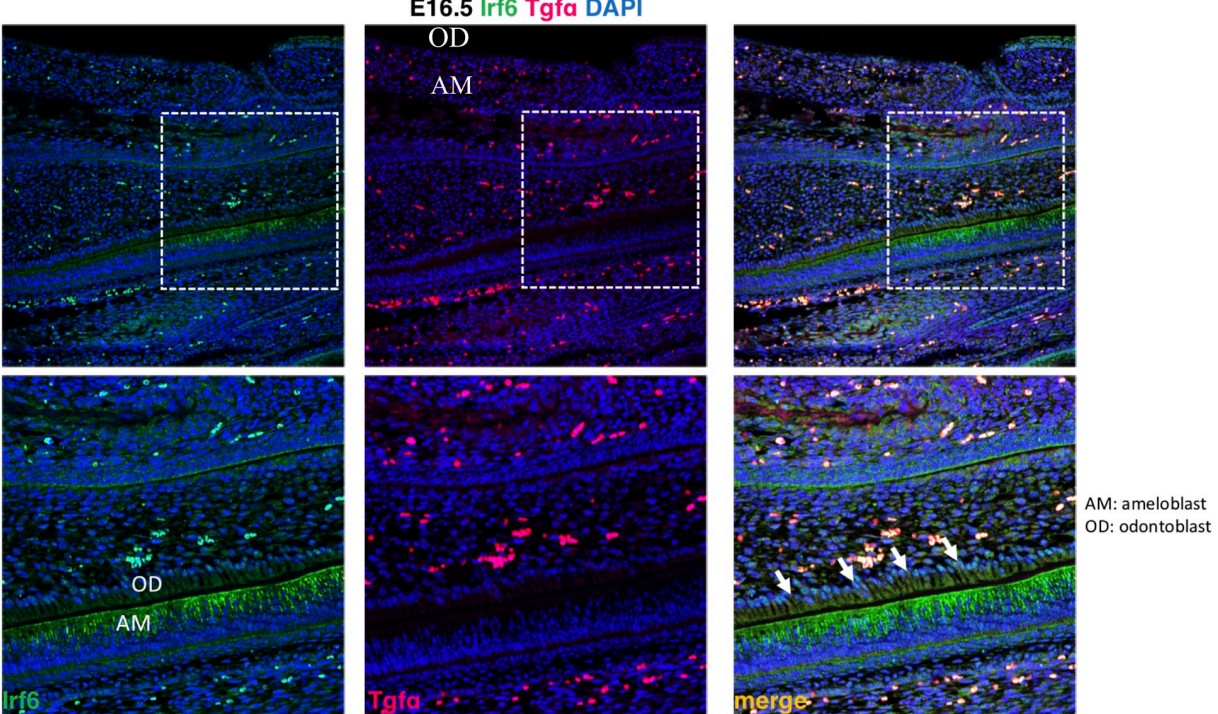

**Fig 1. Expression of Irf6 and Tgfa in sagittal sections of E16.5 wild type murine embryos.** Irf6 is expressed in ameloblasts (AM) but not odontoblasts (OD), whereas Tgfa expression was not detected.

evidence for an interaction of *IRF6* and *TGFA* genotypes and medication intake at 3 years of age (Table 4).

## Discussion

MIH was originally defined as hypomineralization of systemic origin of one to four permanent first molars frequently associated with affected incisors [38]. This definition influenced the work done for more than a decade that focused on the identification of an environmental etiological factor [27]. Our group brought to light the possibility that MIH, in fact, has a multifactorial mode of inheritance [2, 5], and therefore has a genetic component, which was estimated in a proportion explaining the variation in the population of at least 20% based on data from twins [39]. Further, our group has proposed that MIH is possibly a localized and multifactorial expression of amelogenesis imperfecta [40].

**Table 4. Summary of gene-environment interaction results considering medications (taken after three years of age) and its association with MIH phenotype in the cohort from Curitiba (Federal University of Paraná) and Rio de Janeiro.**

| Gene | SNP | Allele | Odds Ratio (95% confidence interval) | p-value with the genetic influence | p-value of medications and MIH only |
|------|-----|--------|--------------------------------------|-------------------------------------|-------------------------------------|
| \multicolumn{6}{Sample from Curitiba (Federal University of Paraná, Brazil)} | | | | | |
| IRF6 | rs17015215 | T | 2.49 (1.31–4.69) | 0.028 | 0.22 |
|      | rs2013162 | A | 2.33 (1.29–4.38) | 0.04 | |
|      | rs861019 | G | 2.5 (1.29–4.72) | 0.026 | |
|      | rs2073487 | C | 2.34 (1.29–4.39) | 0.04 | |
|      | rs642961 | A | 2.36 (1.28–4.44) | 0.037 | |
| TGFA | rs2166975 | A | 2.43 (1.3–4.57) | 0.032 | |
|      | rs1523305 | C | 2.3 (1.3–4.32) | 0.044 | |
|      | rs2902345 | T | 2.4 (1.33–4.47) | 0.038 | |
| Gene | SNP | A1 | Odds Ratio (95% confidence interval) | p-value with the genetic influence | p-value of medications and MIH only |
| \multicolumn{6}{Sample from Rio de Janeiro, Brazil} | | | | | |
| IRF6 | rs17015215 | T | 4.11 (1.81–7.42) | 0.0009 | 0.0002 |
|      | rs2013162 | A | 5.16 (2.43–8.87) | 0.0002 | |
|      | rs861019 | G | 6.86 (3.33–10.4) | 0.0004 | |
|      | rs2073487 | T | 5.74 (2.47–10.02) | 1.9e-005 | |

Some argue that the genetic condition affecting enamel is called amelogenesis imperfecta, and that affect all teeth and have a Mendelian mode of inheritance. On the other hand, enamel hypomineralization (enamel hypoplasia or enamel hypocalcification) would be simply caused by impaired ameloblast function and not by altered genes. Therefore, enamel hypomineralization would be a chronological disturbance that would encompass MIH. This line of thought has led to the hypothesis that the cause of the impaired ameloblast function could be identified because that would be the result of systemic or local causes and this hypothesis has hindered progress of the field for the last 20 years. MIH, like dental caries or periodontitis, fits well in the complex or multifactorial inheritance framework, and similar to cardiovascular diseases as an example, is determined by more than one gene and can be influenced by the environment.

The evidence that MIH frequency varies across the world [6–22] is supported by the results found in the present study. The association found in the Turkish population was different from the results obtained in the cohorts from Brazil, which indeed indicates that genetic factors contributing to the MIH phenotype may vary depending on geographic origin. Both MIH and clefts apparently have higher frequencies in the north of Europe and this frequency declines as one travels toward the Mediterranean [36, 41]. Interestingly enough, these differences are not clearly seen for the figures from the Americas, with Brazil and the United States with reported frequencies of 10% to 13% [16, 19–21]. Additional evidence that interferon regulatory factor 6 *(IRF6)* and transforming growth factor alpha *(TGFA)* may interact, as previously suggested in the formation of cleft lip and palate [33], is also supported here as for the formation of MIH in the Turkish population. It was not possible to determine the exact mechanism underlying this population-specific association. The interpretation of these results should be analyzed cautiously. When one looks at the frequency of the less common allele of rs17015215 in the Turkish sample, there are two thirds of the individuals affected by MIH in comparison to 1/7 of the individuals that served as comparison. The frequency of the less common allele of rs930655 is also low: 9/39 in the affected individuals and 9/54 in the unaffected. When using these small numbers, one obtains very high odds ratios between 25 and 40. The issue of overestimation of odds ratios is well documented in the literature [42–52]. To avoid the

overestimation of the strength of the association, when one wishes odds ratios to be approximations of relative risk, prevalence ratios were calculated, which are the ones presented in Table 3.

Our group has previously reported an interaction of *TGFBR1* and childhood pneumonia as a possible gene-environment mechanism that increases the chance of MIH occurrence. Additionally, it was reported statistical evidence of potential interaction between genes related to tooth development and immune response [29]. This present work reports evidence for both gene-gene (*IRF6-TGFA*) and gene-environment (*IRF6*-medications taken and *TGFA*-medications taken) interactions, which support the hypothesis that MIH is a complex genetic condition [5]. In the gene expression analysis, it was observed expression of Irf6 in ameloblasts, but expression of Tgfa was not observed. These results can possibly be explained by the timing of the analysis or simply meant that *TGFA* and *IRF6* do not colocalize, and hence, have no direct joint effect of enamel disturbances. The variable medication intake at 3 years of age was created to capture children with history of medication intake from birth and past 3 years and to serve as a surrogate for children more prone to infections and/or illnesses. The 52 children that were included in these analyses had a perfect correlation of medication used before 1 year of age and medication used at 3 years of age. Three years of age was the cutoff chosen with the assumption that mothers would more likely remember that phase since their toddler was able to speak approximately 200 words and being mostly understood. Knowing that enamel development of first permanent molars is completed around 4 years of age [23], a possible mechanism underlying MIH is an interference in enamel formation due to an organic stress in response to illness or infection. This organic stress could happen at any time between birth and past 4 years of age in children with a particular genetic background that includes hypomorphic alleles of *IRF6*, *TGFA*, and other genes such as *TGFBR1* [29], and *AQP5* [28]. It is likely that disturbances of first permanent molar development may happen after the first year of life, since mineralization of these teeth is not complete until later.

Previous work suggested that MIH is associated with medication intake although this finding was not confirmed by others [26]. The use of medications, which is a surrogate for illness, apparently can lead to MIH depending on genetic variation in *IRF6* and *TGFA*. To properly confirm this hypothesis, the establishment of a cohort of pregnant women that can be prospectively followed from birth to the timing when the permanent dentition is developing is suggested. This approach would allow to minimize issues of recall bias and to provide more definitive answers that the currently published literature could not.

The *IRF6-TGFA*-medication association with MIH also supports the idea that MIH and cleft lip and palate could possibly be linked. A pleiotropic gene effect could explain this link. Our group has proposed that isolated cleft lip and palate actually is a syndrome that involves disturbances of the dentition [34] and one line of investigation would be to test if genetic variants associated with oral clefts also associate with MIH. We have proposed this approach for isolated tooth agenesis as well [53], and it provided a tool for gene discovery [54, 55]. Tgfa and Irf6 did not show the same pattern of expression in mouse teeth and the mechanism of how they may interact to lead to MIH is yet to be determined.

Among the limitations that can be listed for this present study are the small sample sizes of each independent cohort that may not have allowed the detection of small effect sizes. Depending on the study, clinical information was obtained from reviewing medical records or by the use of questionnaires, and these differences can be source of variation. The fact that detailed information about the type of medication the patients were taking was not available due to self-reporting of this information forced the inclusion of any type of medication intake in the analysis. However, these were not vitamins or other supplements. This

kind of observational study that aims to recover information from many years prior suffers from potential issues related to recall bias. For these reasons, the results presented here should be taken cautiously. Another limitation that can be pointed out is that the question on medications taken was asked years after the fact at the moment of participation in the study. Depending on the cohort studied, assessments were done at different age groups (6 to 12, 6 to 10, or at 8 years of age) and one can think this may introduce some variation. In contrast, the strengths of this study include the well characterized phenotype studied here, determined by experienced dentists from different centers, and the diverse geographic populations studied. The combination of these factors increases the confidence in the results obtained.

In summary, the present study provided evidence that *IRF6* and *TGFA* might interact and be involved with MIH in certain populations, although none of these genes appear to have by themselves a major role in MIH. This effect may become more likely apparent when children have to make use of medications around the age of three years. Additional studies to test potential gene-gene interactions in diverse populations are warranted in order to confirm results reported by previous studies [29] and the results found here. Study designs aiming to obtain a more detailed information about what specific medications are involved in the development of MIH are also suggested. These studies should try minimize recall bias by having precisely defined research questions, appropriate data collection methods, well-trained interviewers, being prospective in nature, blinding for researchers and patients, and including nested case-control designs. Despite the limitations described for the present study, the results both reinforce the detrimental effects that medication intake can cause to oral health, and the link between genes associated with orofacial clefts and the MIH phenotype.

## Methods

### Subjects

The total study population consisted of 1,065 individuals from four populations including three Brazilian cohorts and a Turkish cohort. Salivary sample collection and DNA extraction procedures were described previously [56]. Two groups were from Curitiba, Brazil; the first group consisted of 356 subjects (169 females and 187 males) and the study protocol was approved by the Municipal Department of Education and the Committee for Ethics in Research in Human Health Sciences of the Federal University of Paraná (UFPR; approval no. 1.613.829/2016). The second group from Curitiba consisted of 200 individuals (108 females and 92 males) and the project was approved by the Ethics and Research Committee of the Pontifical Catholic University of Paraná, under the protocol number 1.971.986. The third cohort was from Rio de Janeiro, Brazil, consisted of 174 individuals (73 females and 101 males), and the study protocol was approved by the local Ethics Committee for Research (Hospital Universitário Clementino Fraga Filho–HUCFF/UFRJ–approval protocol number 4598514.7.00005257). Finally, the Turkish population consisted of 335 subjects (164 females and 171 males) and the study was approved by the Istanbul University (approval protocol number 2006/2508) Institutional Review Board. Further oversight was obtained at the University of Pittsburgh Institutional Review Board (IRB approval protocol numbers PRO0710045 and PRO12080056). MIH was evaluated using the European Academy of Pediatric Dentistry (EAPD) criteria [57], and all subjects/guardians read and signed a written informed consent before their participation in the study.

## Cohort from Curitiba, Brazil (Federal University of Paraná): Eligibility criteria, calibration of the examiners and data collection

The sample consisted of 356 school children, which were randomly selected from a representative population [58] from the city of Curitiba public schools. To ensure accurate population representation, two-stage cluster sampling was performed. First, two schools from each administrative district were randomly selected. Next, two to three classrooms from each selected school were randomly chosen using the website www.randomizer.org. The children included were all eight years old and presenting four erupted first molars in the oral cavity. Subjects who had orthodontic braces, syndromes, or amelogenesis imperfecta were excluded from the study. The affected group consisted of children with at least one tooth affected by MIH and the comparison group consisted of children who had no teeth presenting hypomineralization.

A structured socioeconomic data questionnaire was completed by the children's caregivers. Four examiners, previously trained and calibrated to diagnose Molar-incisor hypomineralization, selected thirty intraoral photographs of MIH affected teeth in order to define the diagnosis. Sixty photographs of different clinical scenarios that involved manifestations of MIH were then analyzed by the examiners. After one week, the examiners independently analyzed the same photographs in a different order (duplicate examination, kappa > 0.75). The clinical examination was performed in a school environment using artificial light, a dental mirror, a dental probe with a blunt tip, and sterile gauze. Data collection was performed from November 2016 to September 2017.

## Cohort from Curitiba, Brazil (Pontifical Catholic University of Paraná): Eligibility criteria, calibration of the examiners and data collection

This sample consisted of a total of 200 individuals, enrolled in Curitiba municipal schools during 2018. Of these, 100 were affected by MIH and 100 non-affected, 108 were females and 92 were males. This cohort was recruited independently from the cohort recruited by investigators at the Federal University of Paraná, however considering the geographical proximity, there is a possibility that some individuals participated in both studies. Although this was a slight possibility, the data from each group were analyzed separately. The inclusion criteria were children between 6 and 10 years of age, affected by MIH in completely erupted teeth. As for exclusion criteria, individuals undergoing orthodontic treatment or scoring 2, 3, 4, 5 and 6 of ICDAS (International Caries Detection and Assessment System) were excluded from the study population. Two interviewers were calibrated on MIH diagnosis using photographs according to a previously published criterion [59]. The intra-rater reliability was performed one month after to the first assessment for the diagnosis of both case and comparison groups (kappa = 1). The clinical examinations were performed by calibrated dentists in the schools, using natural light, an oral mirror and an exploratory probe when it was necessary.

## Cohort from Rio de Janeiro, Brazil: Eligibility criteria, calibration of the examiners and data collection

This sample consisted of 174 individuals, from 7 to 14 (10.13±1.9) years of age, treated at the Pediatric Dentistry Clinic of the Federal University of Rio de Janeiro. Recruitment took place from July 2015 to April 2017, and divided into children with MIH (n = 78) and without MIH (n = 96). The eligibility criteria for this study included patients having all the first permanent molars erupted. The exclusion criteria consisted of children affected by congenital syndromes,

enamel defects (hypoplastic lesions, fluorosis, amelogenesis imperfecta or tetracycline stain) and patients undergoing orthodontic treatment.

To assess for inter and intra examiner-reliability, a theoretical exercise was previously applied for two days. Twenty clinical images of dental enamel defects including fluorosis, hypoplasia, amelogenesis imperfecta, dental caries and MIH were shown to the examiners (F.M.F.S., M.C.M). Two weeks after the first assessment, a new assessment was carried out with the examiners and the kappas were 0.88 and 0.89, respectively.

The subjects' health data were collected through health records, including the demographic data (child´s age, sex, place of birth, residence, parent's education and income), mother's health during pregnancy, and children's medical history (medication intake, systemic diseases, high fever, malnutrition, asthma, bronchitis, epilepsy, and severe infections). The clinical examination was performed by a calibrated dentist (F.M.F.S). Examinations were carried out using a mirror and a probe, at the dental chair and using artificial light.

### Cohort from Istanbul, Turkey: Eligibility criteria, calibration of the examiners and data collection

Eligible individuals were enrolled at the Pediatric Clinic of Istanbul University, Turkey (n = 335, ages ranging from 6 to 12 years). Subjects affected by syndromes, fluorosis, or the ones who had fixed appliances were excluded from the study. Cases were defined as subjects affected by MIH, while controls were defined as subjects with no evidence of MIH.

Calibrated examiners carried out the clinical examination, with F.S. having calibrated M.B. and M.K. Exam calibrations were performed according to the following protocol: First, the calibrator presented to the examiner the criteria for MIH detection, showing pictures of several situations to be observed in the exam and discussing each of these situations in a session that lasted one to two hours. Next, the calibrator and examiner analyzed ten to twenty subjects and discussed each case. The intraexaminer agreement was assessed by a second clinical examination in 10% of the sample after two weeks, with a kappa of 1.0. M.K. pre-screened subjects, and M.B. performed the full exam. This cohort and methodology have been reported before [2]. Clinical examinations were performed using a flashlight and intra-oral mirror and gauze was used to dry and clean the teeth prior to examination. Artificial light and dental operatory were used for all evaluations. An explorer was gently used for assessing the smoothness of tooth surfaces.

### Single marker analysis

The amount of DNA and the purity of each sample was determined by spectrophotometry (NanoDrop 1000, Thermo Fisher Scientific, US). The DNA concentration was obtained by readings at 260 nm and the purity by 260/ 280 nm proportion. Five single nucleotide polymorphisms (SNPs) in *IRF6* (rs2073487, rs2013162, rs17015215, rs861019, rs642961) and four in *TGFA* were genotyped (Table 2). PCR reactions were carried out using Taqman chemistry [60] in 3.0 μl reaction volumes in an ABI PRISM Sequence Detection System 7900 (Applied Biosystems, Foster City, CA, USA). Genotyping calls were analyzed using SDS software version 1.7 (Applied Biosystems). PCR reactions were repeated twice when necessary and allele frequencies were calculated. Genotypes are available as (S1–S4 Files).

### Gene-gene interactions

Gene-gene interaction analyses were performed using logistic regression as implemented in PLINK. In order to test the interaction between *IRF6* and *TGFA*, the presence of MIH was considered a dependent variable and the SNPs were considered independent variables. A file

containing all *IRF6* genotypes was ran against a covariate file containing the *TGFA* genotypes organized by copies of alleles, and this analysis was repeated with a *TGFA* genotype file against the *IRF6* covariate file.

## Immunofluorescence

Expression of Tgfa and Irf6 proteins was performed on paraffin sections from heads of wild type mice at E16.5. Maintenance and handling of mice were approved by the Animal Care Unit at the University of Iowa. Tissues were deparaffinized and rehydrated in a series of ethanol dilutions. Slides were boiled for 20 min in antigen unmasking solution (Vector Laboratories, H-3300). Sections were blocked with 20% donkey serum (sigma, D9663A), then incubated overnight at 4˚C with the following primary antibodies: polyclonal goat anti-Tgfa (1:50, R&D, AF-239-SP) and polyclonal rabbit anti-Irf6 (1:50, sigma SAB2102995). After rinsing in PBS, sections were incubated with secondary antibodies conjugated to Alexa Fluorophore 488 or 555 (Molecular Probes, Invitrogen, CA). The nuclei were counterstained with DAPI in PBS (1:10000). The images were taken using a ZEISS 700 confocal microscopy.

## Gene-environment interactions

Tests for potential associations between MIH and factors that affected the children of 3 years of age or older in the cohorts from Curitiba (Federal University of Paraná) and Rio de Janeiro were carried out. The variables included the presence of respiratory issues, malnutrition, any type of food intolerance, infection of any sort and any type of medication intake between 3 years of age and the time the sample was collected. This information was part of the questionnaires obtained for both MIH affected and unaffected children. Mothers responded the questions and all participants were asked the same questions, which included medical history of the child (diseases and medications taken) and details of the child's diet. Vitamins and other supplements were not counted as medications.

The variable "medications taken at 3 years of age" was designed due to the evidence that cuspal enamel formation of first permanent molars is not completed until past 3 years of age [24]. These data suggest that the first molar susceptibility window for MIH goes beyond the first year of age. The suggestion that first permanent molar mineralization is completed by one year of age likely limits the ability of unveiling associations.

We have complied with the STROBE guidelines in this study.

## Statistical analysis

All SNPs in all samples were tested for deviation from Hardy–Weinberg equilibrium using chi-square. In the single-marker analysis, association tests were performed comparing genotypes to phenotype between affected individuals and their respective comparison groups as implemented in the PLINK software [61]. Both odds ratios and 95% confidence intervals were calculated. In order to account for multiple testing, Bonferroni correction was performed and p-values of 0.001 (0.05/36) or below were considered significant.

In the gene-gene interaction analysis, p-values below 0.0004 (0.05/108) were considered statistically significant.

In the gene-environment interaction analysis the covariates were dichotomous and analyses were performed using logistic regression as also implemented in PLINK.

The power necessary to detect significant associations between a gene marker and the phenotype in each cohort was calculated using a Genetic Power Calculator tool [62]. The frequency of the high-risk allele in the population was defined as equal to 0.2, the prevalence of MIH as 0.12, D' as 1.0 (amount of linkage disequilibrium), the respective number of cases of

each cohort and the case-comparison group ratio. When imputing the effect size in the heterozygote form as 1.3 and the effect size of homozygous form as 5.0, it was obtained 69%, 57%, 51%, and 79% power in the cohort from Curitiba (Federal University of Paraná), Curitiba (Pontifical Catholic University), Rio de Janeiro and Istanbul, respectively. Power will decrease with lower effect sizes for both forms, and will increase to 82%, 70%, 64%, and 89%, respectively if the effect size for the homozygous form is 6.0.

## Supporting information

**S1 File. Raw genotypes, Curitiba (UFPR).**
(XLSX)

**S2 File. Raw genotypes, Curitiba (PUCPR).**
(XLSX)

**S3 File. Raw genotypes, Rio de Janeiro.**
(XLSX)

**S4 File. Raw genotypes, Istanbul.**
(XLSX)

## Acknowledgments

This paper is based in part on a thesis submitted to the graduate faculty, Federal University of Rio de Janeiro, in partial fulfillment of the requirements for the PhD degree (for F.M.F.S.) and on a thesis submitted to the graduate faculty, Pontifical Catholic University of Paraná, in partial fulfillment of the requirements for the MS degree (for E.G.C.).

## Author Contributions

**Conceptualization:** Mariana Bezamat, Ariadne Letra, Figen Seymen, Brad Amendt, Renata I. Werneck, Alexandre R. Vieira.

**Formal analysis:** Alexandre R. Vieira.

**Funding acquisition:** Alexandre R. Vieira.

**Investigation:** Mariana Bezamat, Juliana F. Souza, Fernanda M. F. Silva, Emilly G. Corrêa, Aluhê L. Fatturi, João A. Brancher, Flávia M. Carvalho, Tayla Cavallari, Laís Bertolazo, Cleber Machado-Souza, Mine Koruyucu, Merve Bayram, Andrea Racic, Benjamin M. Harrison, Yan Y. Sweat, Deborah Studen-Pavlovich, Figen Seymen, Brad Amendt, Renata I. Werneck, Marcelo C. Costa, Adriana Modesto.

**Methodology:** Juliana F. Souza, Ariadne Letra, Figen Seymen, Brad Amendt, Renata I. Werneck, Marcelo C. Costa, Adriana Modesto, Alexandre R. Vieira.

**Resources:** Alexandre R. Vieira.

**Software:** Mariana Bezamat, Benjamin M. Harrison.

**Supervision:** Marcelo C. Costa, Adriana Modesto, Alexandre R. Vieira.

**Writing – original draft:** Mariana Bezamat, Alexandre R. Vieira.

**Writing – review & editing:** Juliana F. Souza, Fernanda M. F. Silva, Emilly G. Corrêa, Aluhê L. Fatturi, João A. Brancher, Flávia M. Carvalho, Tayla Cavallari, Laís Bertolazo, Cleber Machado-Souza, Mine Koruyucu, Merve Bayram, Andrea Racic, Benjamin M. Harrison,

Yan Y. Sweat, Ariadne Letra, Deborah Studen-Pavlovich, Figen Seymen, Brad Amendt, Renata I. Werneck, Marcelo C. Costa, Adriana Modesto.

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
