## [Decision Letter · Decision Letter 0]

9 Sep 2020

PONE-D-20-24540

Gene-environment interaction in molar-incisor hypomineralization

PLOS ONE

Dear Dr. Vieira,

Thank you for submitting your manuscript to PLOS ONE. After careful consideration, we feel that it has merit but does not fully meet PLOS ONE’s publication criteria as it currently stands. Therefore, we invite you to submit a revised version of the manuscript that addresses the points raised during the review process.

The expert reviewers have identified several major issues. Of specific concern is the description of variables, potential missing data, inclusion criteria, approach to analysis and data interpretation.

We look forward to receiving your revised manuscript.

Kind regards,

JJ Cray Jr., Ph.D.

Academic Editor

PLOS ONE

Journal Requirements:

Reviewers' comments:

Reviewer's Responses to Questions

**Comments to the Author**

1. Is the manuscript technically sound, and do the data support the conclusions?

Reviewer #1: No

Reviewer #2: Yes

Reviewer #3: Partly

Reviewer #4: No

2. Has the statistical analysis been performed appropriately and rigorously? 

Reviewer #1: Yes

Reviewer #2: Yes

Reviewer #3: I Don't Know

Reviewer #4: No

3. Have the authors made all data underlying the findings in their manuscript fully available?

Reviewer #1: Yes

Reviewer #2: Yes

Reviewer #3: No

Reviewer #4: Yes

4. Is the manuscript presented in an intelligible fashion and written in standard English?

Reviewer #1: Yes

Reviewer #2: No

Reviewer #3: Yes

Reviewer #4: Yes

5. Review Comments to the Author

Reviewer #1: In this study, the authors analyzed causes of Molar Incisor Hypomineralization (MIH). The term “MIH” came into use 20 years ago, and causes of MIH are unknown. This reviewer thinks that the topic is very important for dentists. However, the following points should be clarified prior to further consideration of publication of the manuscript.

MIH is defined as a hypomineralization of systemic origin of one to four permanent first molars frequently associated with affected incisors by Weerheijm (2001).

Ref: Molar incisor hypomineralisation (MIH) Eur J Paediatr Dent. 2003;4(3):114-20.

The authors described that these enamel defects usually occur when there are disturbances during the mineralization or maturation stage of amelogenesis. The first molar is finished mineralization of crown by 3 years old. However, the authors concluded that environmental factors affecting children that were 3 years of age or older were also hypothesized to play a role in the disease etiology.

The authors also described that genes interacted and contributed to predisposition of MIH. If genetic factors cause hypomineralization, the disease is not MIH but amelogenesis imperfecta. Genetic abnormalities cause hypoplasia in all teeth rather than locally.

Reviewer #2: The English language needs to be professionally revised. There are straightforward grammar issues. For example paragraph 4 in introduction “Risk factors for MIH did not fully explained” … and paragraph 5 in introduction “has been documented as responsible for not only affect amelogenesis…”. “The chance of the less common allele of TGFA rs930655 in addition to the IRF6 marker allele to increase the chance …”

Reviewer #3: This manuscript deals with an interesting topic and currently one of the forefront problems in the field of paediatric dentistry, i.e. MIH, focusing on its aetiology. However, there are some issues with this manuscript. In comments, only methodology and results of the study are addressed.

There is no information on how children were selected. Data on children’s age is missing.

Who performed dental examinations? Were the dentists who performed dental examination calibrated?

Regarding MIH diagnostic criteria; description of the dental examination results related to MIH affected teeth is essential (e.g. which teeth are MIH-affected, what was MIH severity).

Text describing environmental factors is very vague. How was the data on environmental factors obtained (e.g. medical history, a questioner)? Who answered the questions about environmental factors (e.g., children, parents, teachers,..)? Were all participants asked the same questions? On which environmental factors that could be associated with MIH were participants asked?

Why do the authors proceed from a hypothesis that environmental factors affecting 3 year-old children or older play a role in the aetiology of MIH? In MIH, the insult to the ameloblasts is likely to occur either prenatally or in the first year of life (Mangum and Farah).

Since the aim of the study was to clarify the aetiology of MIH, it is not clear why various potentially harmful aetiological factors would be considered as a single potential cause of MIH.

The results of the study did not show significant association between the selected SNPs and MIH. The study also does not provide solid evidence for the IRF6 and TGFA interaction, nor that the development of MIH is associated with any of these genes. Moreover, as the authors have already stated themselves, the size of each cohort is small for such research.

Figure 1: A description of each of the six images is missing.

Reviewer #4: Thank you for the opportunity to review this manuscript, ‘Gene-environment interaction in Molar Incisor Hypomineralisation’. I congratulate the authors for undertaking very broad and comprehensive approach to investigating genetic interactions in MIH. However, due to two major concerns with this paper, I cannot support publication of this manuscript. My main concerns are:

(1) The choice of genetic variants – these appear to have been measured due to their relevance in a different condition (cleft lip and palate) which the authors suggest may be linked. However, adopting this candidate-based approach and then, by stratifying the data when significant results were not obtained so that a ‘statistically significant’ could be reported, seems to misrepresent the real outcome of the study. I suggest there are potentially more biological plausible interactions that could have been investigated that could be supported by existing evidence relating potential genetic and environmental risk factors. I note the authors do acknowledge some of the limitations however these major limitations do not seem to be considered when reporting the study conclusions.

(2) The observational component is particularly weak – participant recall of ‘medication use’ is prone to many biases and does not contribute to the existing evidence base regarding MIH. In order to investigate a complex causal relationship such as this, the authors needed to develop a detailed analysis plan with consideration of potential confounders etc.

6. PLOS authors have the option to publish the peer review history of their article (what does this mean?). If published, this will include your full peer review and any attached files.

Reviewer #1: No

Reviewer #2: **Yes: **Azza Tagelsir Ahmed

Reviewer #3: No

Reviewer #4: No

---

## [Author Response · Author response to Decision Letter 0]

11 Sep 2020

September 9, 2020

To: The Editor, PLoS One

Dear Dr. Cray Jr.:

after carefully reading yours and the reviewers’ comments, we decided to substantially revise and resubmit our work entitled “Gene-environment interaction in molar-incisor hypomineralization .” We believe we have addressed all the concerns and are hoping it is suitable for publication in your journal.

This work explores genomic variants associated with cleft lip and palate as potentially associated with MIH.

We are looking forward to seeing our revised work being well received. Below are point-by-point answers for the critiques and all changes are marked in yellow.

Sincerely,

Alexandre R. Vieira, D.D.S., M.S., PhD

Professor of Oral Biology, Pediatric Dentistry and Human Genetics Director of Clinical Research and Director of Student Research Department of Oral Biology

University of Pittsburgh School of Dental Medicine

412 Salk Pavilion

Pittsburgh, PA 15213

Office: 412-383-8972

FAX: 412-624-3080

E-mail: arv11@pitt.edu

Point-by-point response to reviewers:

Reviewer #1: In this study, the authors analyzed causes of Molar Incisor Hypomineralization (MIH). The term “MIH” came into use 20 years ago, and causes of MIH are unknown. This reviewer thinks that the topic is very important for dentists. However, the following points should be clarified prior to further consideration of publication of the manuscript.

MIH is defined as a hypomineralization of systemic origin of one to four permanent first molars frequently associated with affected incisors by Weerheijm (2001).

Ref: Molar incisor hypomineralisation (MIH) Eur J Paediatr Dent. 2003;4(3):114-20.

The authors described that these enamel defects usually occur when there are disturbances during the mineralization or maturation stage of amelogenesis. The first molar is finished mineralization of crown by 3 years old. However, the authors concluded that environmental factors affecting children that were 3 years of age or older were also hypothesized to play a role in the disease etiology.

The authors also described that genes interacted and contributed to predisposition of MIH. If genetic factors cause hypomineralization, the disease is not MIH but amelogenesis imperfecta. Genetic abnormalities cause hypoplasia in all teeth rather than locally.

RESPONSE: The definition cited above from Karin Weerheijm (2001) that the condition is o systemic origin does not exclude the possibility that it has a genetic component. We have suggested this for 8 years now and more and more scientists have surrendered to the idea at this point. We indeed believe MIH is possibly an extension of amelogenesis imperfecta, a localized one, at least in some instances. We added the opening paragraph of the Discussion section to address this reviewer’s concern. Further, we had originally included discussion in the paper regarding the time of mineralization of molars crowns and the variable “infection at 3 years of age.” It was on the original third paragraph of the Discussion section.

Reviewer #2: The English language needs to be professionally revised. There are straightforward grammar issues. For example paragraph 4 in introduction “Risk factors for MIH did not fully explained” … and paragraph 5 in introduction “has been documented as responsible for not only affect amelogenesis…”. “The chance of the less common allele of TGFA rs930655 in addition to the IRF6 marker allele to increase the chance …”

RESPONSE: We carefully revised the text for grammar and style and made the above pointed out corrections.

Reviewer #3: This manuscript deals with an interesting topic and currently one of the forefront problems in the field of paediatric dentistry, i.e. MIH, focusing on its aetiology. However, there are some issues with this manuscript. In comments, only methodology and results of the study are addressed.

There is no information on how children were selected. Data on children’s age is missing.

RESPONSE: This information was in the supplemental file. We moved if to the text.

Who performed dental examinations? Were the dentists who performed dental examination calibrated?

RESPONSE: This information was in the supplemental file. We moved if to the text.

Regarding MIH diagnostic criteria; description of the dental examination results related to MIH affected teeth is essential (e.g. which teeth are MIH-affected, what was MIH severity).

RESPONSE: This information was in the supplemental file. We moved if to the text. The original cohorts had originally assigned subjects are affected or not affected.

Text describing environmental factors is very vague. How was the data on environmental factors obtained (e.g. medical history, a questioner)? Who answered the questions about environmental factors (e.g., children, parents, teachers,..)? Were all participants asked the same questions? On which environmental factors that could be associated with MIH were participants asked?

RESPONSE: We added the information as requested. How the data was originally obtained (questionnaire) was original described in the methods.

Why do the authors proceed from a hypothesis that environmental factors affecting 3 year-old children or older play a role in the aetiology of MIH? In MIH, the insult to the ameloblasts is likely to occur either prenatally or in the first year of life (Mangum and Farah).

RESPONSE: We had originally added some Discussion regarding the variable of infection at 3 years of age. It is quite possible the disruption of amelogenesis happens later as well, not necessarily during the secretion phase only, but during the mineralization phase as well.

Since the aim of the study was to clarify the aetiology of MIH, it is not clear why various potentially harmful aetiological factors would be considered as a single potential cause of MIH.

RESPONSE: It is not the case we considered the various risk factors as a single potential cause. It is just that our analysis suggest one particular one associated. 

The results of the study did not show significant association between the selected SNPs and MIH. The study also does not provide solid evidence for the IRF6 and TGFA interaction, nor that the development of MIH is associated with any of these genes. Moreover, as the authors have already stated themselves, the size of each cohort is small for such research.

RESPONSE: This is correct and it is clearly stated in the Results and Discussion sections. Suggestive results only come from the combined analyses of different factors.

Figure 1: A description of each of the six images is missing.

RESPONSE We added the description as requested.

Reviewer #4: Thank you for the opportunity to review this manuscript, ‘Gene-environment interaction in Molar Incisor Hypomineralisation’. I congratulate the authors for undertaking very broad and comprehensive approach to investigating genetic interactions in MIH. However, due to two major concerns with this paper, I cannot support publication of this manuscript. My main concerns are:

(1) The choice of genetic variants – these appear to have been measured due to their relevance in a different condition (cleft lip and palate) which the authors suggest may be linked. However, adopting this candidate-based approach and then, by stratifying the data when significant results were not obtained so that a ‘statistically significant’ could be reported, seems to misrepresent the real outcome of the study. I suggest there are potentially more biological plausible interactions that could have been investigated that could be supported by existing evidence relating potential genetic and environmental risk factors. I note the authors do acknowledge some of the limitations however these major limitations do not seem to be considered when reporting the study conclusions.

RESPONSE: We revised the conclusions to address this concern.

(2) The observational component is particularly weak – participant recall of ‘medication use’ is prone to many biases and does not contribute to the existing evidence base regarding MIH. In order to investigate a complex causal relationship such as this, the authors needed to develop a detailed analysis plan with consideration of potential confounders etc.

RESPONSE: This is an issue inherent to any observational study that includes recalling events many years prior. We added text in the last paragraph of the discussion to acknowledge this.

---

## [Decision Letter · Decision Letter 1]

22 Sep 2020

PONE-D-20-24540R1

Gene-environment interaction in molar-incisor hypomineralization

PLOS ONE

Dear Dr. Vieira,

Thank you for submitting your manuscript to PLOS ONE. After careful consideration, we feel that it has merit but does not fully meet PLOS ONE’s publication criteria as it currently stands. Therefore, we invite you to submit a revised version of the manuscript that addresses the points raised during the review process.

Although the manuscript was given an overhaul, the same themes concerning major issues were again expressed.

We look forward to receiving your revised manuscript.

Kind regards,

JJ Cray Jr., Ph.D.

Academic Editor

PLOS ONE

Reviewers' comments:

Reviewer's Responses to Questions

**Comments to the Author**

1. If the authors have adequately addressed your comments raised in a previous round of review and you feel that this manuscript is now acceptable for publication, you may indicate that here to bypass the “Comments to the Author” section, enter your conflict of interest statement in the “Confidential to Editor” section, and submit your "Accept" recommendation.

Reviewer #1: (No Response)

Reviewer #2: (No Response)

2. Is the manuscript technically sound, and do the data support the conclusions?

Reviewer #1: No

Reviewer #2: Yes

3. Has the statistical analysis been performed appropriately and rigorously? 

Reviewer #1: No

Reviewer #2: N/A

4. Have the authors made all data underlying the findings in their manuscript fully available?

Reviewer #1: Yes

Reviewer #2: Yes

5. Is the manuscript presented in an intelligible fashion and written in standard English?

Reviewer #1: Yes

Reviewer #2: No

6. Review Comments to the Author

Reviewer #1: This author commented two problems of this manuscript. I couldn't get satisfying answers from authors.

The causes of disorders in amelogenesis can be divided into those caused by genes (genetic factors) and those not caused by genes (systemic causes, local causes). The former is hereditary disorder (amelogenesis imperfecta), which affects all teeth. The latter is enamel hypomineralization (enamel hypoplasia or enamel hypocalcification) caused by impaired ameloblast function. Enamel hypomineralization is chronological disturbance. MIH is included in enamel hypomineralization. MIH is a local enamel hypomineralization not amelogenesis imperfecta. Authors responded to my comments as that “MIH is possibility an extension of amelogenesis imperfecta”. The authors also have suggested that genetic abnormality is a one of the factors of MIH. I cannot believe this hypothesis. The disorder is not hypomineralization of enamel but amelogenesis imperfecta. “amelogenesis imperfecta” and “enamel hypomineralization” are often misunderstood as a single disorder and expressed in a term. However, they are actually different disorders. The former means hereditary disorders while the latter means nonhereditary congenital disorders considered as chronological disturbances. If enamel hypomineralization is caused by genetic abnormalities, enamel hypomineralization (MIH) is recognized in all teeth.

This reviewer also commented on the timing of enamel mineralization. MIH is defined hypomineralization of enamel affecting affects one or more permanent first molars with or without permanent incisor involvement. Calcification of the first molar begins at birth, and crown formation (mineralization of enamel) completed at 30-36 months. The same opinion was found in reviewer #3.

Reviewer #3 Why do the authors proceed from a hypothesis that environmental factors affecting 3 year-old children or older play a role in the aetiology of MIH? In MIH, the insult to the ameloblasts is likely to occur either prenatally or in the first year of life.

The authors responded that “It is quite possible the disruption of amelogenesis happens later as well, not necessarily during the secretion phase only, but during the mineralization phase as well.”

This reviewer thinks that the authors don’t consider mineralization phase of the first molar. Environmental factors affecting children that were 3 years of age or older never concern to MIH. This factor affects permanent teeth except for the first molars.

Reviewer #2: After reviewing the revised submission and the response to the reviewer comments, I would like to inform the authors that NOT all my concerns were addressed. only the English language point was addressed!!

Please refer to the attached point by point review document (8 points in total) and provide point by point explanations and amendments within the body of the manuscript.

7. PLOS authors have the option to publish the peer review history of their article (what does this mean?). If published, this will include your full peer review and any attached files.

Reviewer #1: No

Reviewer #2: **Yes: **Azza Tagelsir Ahmed

---

## [Author Response · Author response to Decision Letter 1]

22 Sep 2020

September 22, 2020

To: The Editor, PLoS One

Dear Dr. Cray Jr.:

after carefully reading yours and the reviewers’ comments, we decided to revise again and resubmit our work entitled “Gene-environment interaction in molar-incisor hypomineralization .” We believe we have addressed all the concerns and are hoping it is suitable for publication in your journal. The two most salient comments from one of the reviewers, the idea that MIH does not have a genetic component and the timing of mineralization of the first permanent first molar being at 1 year of age appeared to us not supported by the current knowledge. We addressed these concerns and added more discussion and believe those are a matter of opinion and we urge that PLoS let the readers react and decide how to interpret our work. Hopefully our work will motivate a change in direction for the field with less of an emphasis on association studies of elusive environmental factors and better designed gene-environmental model studies.

This work explores genomic variants associated with cleft lip and palate as potentially associated with MIH.

We are looking forward to seeing our revised work being well received. Below are point-by-point answers for the critiques and all changes are marked in yellow.

Sincerely,

Alexandre R. Vieira, D.D.S., M.S., PhD

Professor of Oral Biology, Pediatric Dentistry and Human Genetics Director of Clinical Research and Director of Student Research Department of Oral Biology

University of Pittsburgh School of Dental Medicine

412 Salk Pavilion

Pittsburgh, PA 15213

Office: 412-383-8972

FAX: 412-624-3080

E-mail: arv11@pitt.edu

Point-by-point response to reviewers:

Reviewer #1: This author commented two problems of this manuscript. I couldn't get satisfying answers from authors.

The causes of disorders in amelogenesis can be divided into those caused by genes (genetic factors) and those not caused by genes (systemic causes, local causes). The former is hereditary disorder (amelogenesis imperfecta), which affects all teeth. The latter is enamel hypomineralization (enamel hypoplasia or enamel hypocalcification) caused by impaired ameloblast function. Enamel hypomineralization is chronological disturbance. MIH is included in enamel hypomineralization. MIH is a local enamel hypomineralization not amelogenesis imperfecta. Authors responded to my comments as that “MIH is possibility an extension of amelogenesis imperfecta”. The authors also have suggested that genetic abnormality is a one of the factors of MIH. I cannot believe this hypothesis. The disorder is not hypomineralization of enamel but amelogenesis imperfecta. “amelogenesis imperfecta” and “enamel hypomineralization” are often misunderstood as a single disorder and expressed in a term. However, they are actually different disorders. The former means hereditary disorders while the latter means nonhereditary congenital disorders considered as chronological disturbances. If enamel hypomineralization is caused by genetic abnormalities, enamel hypomineralization (MIH) is recognized in all teeth.

RESPONSE: The assumption that a genetic condition affecting teeth would have to be present in al teeth is simply incorrect. We understand that is a misconception in the field and we added more discussion to the matter to address this concern. It is not a matter of opinion anymore, MIH fits well a condition with a complex mode of inheritance. Genetic conditions that have mendelian forms of inheritance are actually not the most typical examples of disease in human. Most common diseases in humans, such as cardiovascular diseases, diabetes, cancer (and dental caries, periodontitis, MIH) have complex modes of inheritance.

This reviewer also commented on the timing of enamel mineralization. MIH is defined hypomineralization of enamel affecting affects one or more permanent first molars with or without permanent incisor involvement. Calcification of the first molar begins at birth, and crown formation (mineralization of enamel) completed at 30-36 months. The same opinion was found in reviewer #3.

RESPONSE: As we discussed originally, most of the children included in the analysis had medications since their first year of life. We described evidence (original reference 19) that the mineralization of first permanent molars can be still occurring pass 36 months and up to 48 months. We added more discussion to highlight this concern.

Reviewer #3 Why do the authors proceed from a hypothesis that environmental factors affecting 3 year-old children or older play a role in the aetiology of MIH? In MIH, the insult to the ameloblasts is likely to occur either prenatally or in the first year of life.

The authors responded that “It is quite possible the disruption of amelogenesis happens later as well, not necessarily during the secretion phase only, but during the mineralization phase as well.”

This reviewer thinks that the authors don’t consider mineralization phase of the first molar. Environmental factors affecting children that were 3 years of age or older never concern to MIH. This factor affects permanent teeth except for the first molars.

RESPONSE: As we discussed originally, most of the children included in the analysis had medications since their first year of life. The impression that first molars are somehow immune from mineralization disturbances after the first year of life, knowing that mineralization of these teeth continues into 36 to 48 months is probably not current. We added more discussion to highlight this concern.

Reviewer #2: After reviewing the revised submission and the response to the reviewer comments, I would like to inform the authors that NOT all my concerns were addressed. only the English language point was addressed!!

Please refer to the attached point by point review document (8 points in total) and provide point by point explanations and amendments within the body of the manuscript.

Throughout the manuscript 

1. English language needs to be professionally revised. There are straightforward grammar issues. For example, paragraph 4 in introduction “Risk factors for MIH did not fully explained” … and paragraph 5 in introduction “has been documented as responsible for not only affect amelogenesis…”. “The chance of the less common allele of TGFA rs930655 in addition to the IRF6 marker allele to increase the chance …”

RESPONSE: We made these corrections in the past version.

2. A disorder is used commonly to refer to a group of conditions. The authors need to logically justify why they are using the word “disorder” to refer to MIH in many areas of the manuscript? Otherwise, the word disorder needs to be substituted throughout.

RESPONSE: We revised the text as requested.

Introduction 3. First paragraph, line 7: Did the authors omit the preposition “with” mistakenly in the sentence “usually permanent incisors”…??? This part of the sentence needs to be rephrased as the involvement of incisors is not always necessary to identify a case of MIH!

RESPONSE: We revised the sentence as suggested.

4. Third paragraph, line 17: “When full maturation is not complete at eruption, a post eruption maturation occurs through mineral ions from the saliva …” All newly eruptive enamel undergoes post eruptive mineralization regardless of the mineralization status of enamel. Please explain why you introduced “post-eruptive mineralization” here? Does it serve any purpose related to the objective of the study? if yes, what?

RESPONSE: We deleted the sentence as suggested

Methods 5. Were the subjects’ medication history collected at 3 years, after 3 years of age, or from birth to 3 years? There are many contradictory and confusing statements. In the beginning of the results section under environmental factors “any type of medications taken after three years of age (Table 4)…” , At the end of the same paragraph “..In both groups we found statistical evidence for an interaction of IRF6 and TGFA genotypes and medication intake at 3 years of age (Table 4)…” Other statements in the discussion section: “The variable medication intake after 3 years of age was created to capture children with history of medication intake from birth and past 3 years….” “another limitation we can point out is that we included only medication taken after 3 years of age…” In the method section: “..any type of medication intake between 3 years of age and the time the sample was collected…” These statements are very confusing and need to be explained? the authors need to be consistent when stating the one (or many) time point(s) where history of medications intake was collected. Please amend and maintain consistency throughout the manuscript.

RESPONSE: We revised the text to make it consistent as suggested. The variable was created to captured use of medication from birth and passed 3 years of age.

6. Referring to the previous point and acknowledging that the mineralization of the first permanent molar takes place from birth up to 3-4 years, why did the authors set up the cut off age to after 3 years to collect medication history? please give detailed explanation.

RESPONSE: 3 years of age is likely easier to remember because the child has achieved a vocabulary of 200 words and most pf what she says can be understood. We added a comment in the discussion to address this concern.

7. What was the reason behind the limited availability of the medication’s intake history to two cohorts out of the four cohorts?

RESPONSE: This variable was not included in these original studies.

8. Referring to the supplemental document, the subjects’ health data were collected from health records in the Rio de Janeiro, Brazil cohort, while structured data questionnaires were used to collect the same data from the Federal University of Paraná cohort. Explain the following issue: a. What do the authors think about the non- homogenous method of collecting medication history between cohorts (questionnaire in one cohort and medical records in another cohort)? this needs to be emphasized in the limitation paragraph in the discussion. How would the previous limitation affect the results of this study? Discussion § Paragraph 1, line 4: the author wanted to emphasize the geographical variations of MIH estimates, however, the generalizability of the declining trend between north and south MIH estimates needs to be addressed with caution!, first of all because the study includes cohorts from south America (Brazil) and the south European/Mediterranean regions. Second, this “claimed” trend is actually reversed when the south American MIH prevalence estimates is compared with the recently published north American MIH estimates (10% - 13% ). Thus, the statement needs to be rephrased to reflect the populations studied!

RESPONSE: We added the suggested limitation on how the data were collected depending on the cohort. The best estimated frequency is Brazil is 13% as well and aligns of the figures from North America. We added a comment in the discussion to acknowledge this reviewer‘s comment.

---

## [Decision Letter · Decision Letter 2]

21 Oct 2020

PONE-D-20-24540R2

Gene-environment interaction in molar-incisor hypomineralization

PLOS ONE

Dear Dr. Vieira,

Thank you for submitting your manuscript to PLOS ONE. After careful consideration, we feel that it has merit but does not fully meet PLOS ONE’s publication criteria as it currently stands. Therefore, we invite you to submit a revised version of the manuscript that addresses the points raised during the review process.

There is one particular concern that should be addressed about the patient medication history. Also if the authors could do another run through for grammar and readibility.

We look forward to receiving your revised manuscript.

Kind regards,

JJ Cray Jr., Ph.D.

Academic Editor

PLOS ONE

Reviewers' comments:

Reviewer's Responses to Questions

**Comments to the Author**

1. If the authors have adequately addressed your comments raised in a previous round of review and you feel that this manuscript is now acceptable for publication, you may indicate that here to bypass the “Comments to the Author” section, enter your conflict of interest statement in the “Confidential to Editor” section, and submit your "Accept" recommendation.

Reviewer #2: (No Response)

2. Is the manuscript technically sound, and do the data support the conclusions?

Reviewer #2: Partly

3. Has the statistical analysis been performed appropriately and rigorously? 

Reviewer #2: N/A

4. Have the authors made all data underlying the findings in their manuscript fully available?

Reviewer #2: Yes

5. Is the manuscript presented in an intelligible fashion and written in standard English?

Reviewer #2: No

6. Review Comments to the Author

Reviewer #2: 1. The English language still needs to be revised. There are many grammar mistakes.

2. The authors response to point 6 where the reviewer is questioning the cut off of 3 years to collect medication history is not satisfying nor convincing.

How is the child's vocabulary status be a valid reason to choose this cut off age, when we know the the medication history is collected through direct questioning of the parent/caregiver?

3. Many sentences need to be re-written in a third person narrative

This sentence is one example but this style needs to be adopted through the manuscript.

"The fact that we did not always have detailed information about the type of medication the patients were taking due to self-reporting of this information forced us to include any type of medication intake in the

analysis. However, these were not vitamins or other supplements. But we are aware that this kind of observational study that aims to recover information from many years prior suffers from potential issues related to recall bias. For these reasons, we ask that our results are taken cautiously.”

7. PLOS authors have the option to publish the peer review history of their article (what does this mean?). If published, this will include your full peer review and any attached files.

Reviewer #2: **Yes: **Azza Tagelsir Ahmed

---

## [Author Response · Author response to Decision Letter 2]

21 Oct 2020

Point-by-point response to reviewer 2:

Reviewer #2: 1. The English language still needs to be revised. There are many grammar mistakes.

RESPONSE: We carefully revised the text for grammar and typos. We did not find many grammar mistakes but corrected the ones identified.

2. The authors response to point 6 where the reviewer is questioning the cut off of 3 years to collect medication history is not satisfying nor convincing.

How is the child's vocabulary status be a valid reason to choose this cut off age, when we know the the medication history is collected through direct questioning of the parent/caregiver?

RESPONSE: We added more detail in the methods to address this concern. Evidence of cuspal enamel formation of first permanent molars can be detected past 3 years of age.

The comment of the vocabulary was in regard to recall bias, not enamel formation.

3. Many sentences need to be re-written in a third person narrative

This sentence is one example but this style needs to be adopted through the manuscript.

"The fact that we did not always have detailed information about the type of medication the patients were taking due to self-reporting of this information forced us to include any type of medication intake in the

analysis. However, these were not vitamins or other supplements. But we are aware that this kind of observational study that aims to recover information from many years prior suffers from potential issues related to recall bias. For these reasons, we ask that our results are taken cautiously.”

RESPONSE: We made the changes to adequate to the style the reviewer is requesting.

---

## [Editor Report · Decision Letter 3]

23 Oct 2020

Gene-environment interaction in molar-incisor hypomineralization

PONE-D-20-24540R3

Dear Dr. Vieira,

We’re pleased to inform you that your manuscript has been judged scientifically suitable for publication and will be formally accepted for publication once it meets all outstanding technical requirements.

Kind regards,

JJ Cray Jr., Ph.D.

Academic Editor

PLOS ONE
---

## [Editor Report · Acceptance letter]

26 Oct 2020

PONE-D-20-24540R3 

Gene-environment interaction in molar-incisor hypomineralization 

Dear Dr. Vieira:

I'm pleased to inform you that your manuscript has been deemed suitable for publication in PLOS ONE. Congratulations! Your manuscript is now with our production department. 

Kind regards, 

on behalf of

Dr. JJ Cray Jr. 

Academic Editor

PLOS ONE